# Importance of Autophagy Regulation in Glioblastoma with Temozolomide Resistance

**DOI:** 10.3390/cells13161332

**Published:** 2024-08-11

**Authors:** Young Keun Hwang, Dong-Hun Lee, Eun Chae Lee, Jae Sang Oh

**Affiliations:** 1Department of Medical Life Sciences, College of Medicine, The Catholic University of Korea, Seoul 06591, Republic of Korea; hyg3954@naver.com (Y.K.H.); lec9589@gmail.com (E.C.L.); 2Industry-Academic Cooperation Foundation, The Catholic University of Korea, 222, Banpo-daro, Seocho-gu, Seoul 06591, Republic of Korea; madeby58@gmail.com; 3Department of Neurosurgery, Uijeongbu St. Mary’s Hospital, College of Medicine, The Catholic University of Korea, Seoul 06591, Republic of Korea

**Keywords:** glioblastoma, autophagy, temozolomide, chemoresistance

## Abstract

Glioblastoma (GBM) is the most aggressive and common malignant and CNS tumor, accounting for 47.7% of total cases. Glioblastoma has an incidence rate of 3.21 cases per 100,000 people. The regulation of autophagy, a conserved cellular process involved in the degradation and recycling of cellular components, has been found to play an important role in GBM pathogenesis and response to therapy. Autophagy plays a dual role in promoting tumor survival and apoptosis, and here we discuss the complex interplay between autophagy and GBM. We summarize the mechanisms underlying autophagy dysregulation in GBM, including PI3K/AKT/mTOR signaling, which is most active in brain tumors, and EGFR and mutant EGFRvIII. We also review potential therapeutic strategies that target autophagy for the treatment of GBM, such as autophagy inhibitors used in combination with the standard of care, TMZ. We discuss our current understanding of how autophagy is involved in TMZ resistance and its role in glioblastoma development and survival.

## 1. Introduction

Glioblastoma (GBM) is one of the most malignant tumors that occur in the central nervous system (CNS) and one of the cancers with a very poor prognosis. According to the World Health Organization (WHO) classification, gliomas are divided into low-grade (I,II) and high-grade (III,IV). While low-grade astrocytomas and rhabdoid glioblastomas have a relatively good prognosis, high-grade anaplastic astrocytomas and high-grade gliomas have a median survival time of only about 15 months [1,2]. According to a recent presentation by the American Association of Neurological Surgeons (AANS), GBMs account for about 54% of all CNS tumors, with an incidence of 3.21 per 100,000 people, an average age of diagnosis of 64 years, more commonly occurring in men compared to women, and a poor survival rate of about 40% in the first year after diagnosis and about 17% in the second year. It is typically diagnosed with CT [3], and intraoperative MRI [4] is also used to guide biopsy [5] and tumor removal. According to the AANS, GBMs commonly develop in the cerebral hemispheres [6], most commonly in the temporal lobe. The main symptoms include headaches, seizures, and neurological deficits. GBMs occur more frequently in adults than meningiomas, the second most common CNS tumor, while medulloblastomas are more common in children. Molecularly, meningiomas typically have deletions of chromosome 22q [7], whereas GBMs have common genetic mutations in the signaling pathways of EGFR, PTEN, and PI3K/AKT/mTOR [8,9].

Multiple signaling pathways in GBMs include amplification of the normal epidermal growth factor receptor (EGFR) and mutations of EGFR, EGFRvIII [9], and PI3K/AKT/mTOR [10]. Expression of EGFR’s mutant EGFRvIII is frequently found in glioma cells and is associated with cell proliferation, tumor growth, treatment resistance, and poor prognosis. PI3K/AKT/mTOR is one of the most important pathways, regulating cell growth, motility, angiogenesis, and metabolism. Dysregulation of PI3K/AKT/mTOR occurs in a majority of GBM tumors and requires regulation [11,12].

Autophagy is an important physiological process that occurs within cells. It involves the cell breaking down and recycling its own damaged proteins, organelles, and other unwanted materials. Autophagy is divided into three main types, which are macroautophagy, chaperone-mediated autophagy, and microautophagy. Macroautophagy, also known as autophagy, is an important autolysis process present in all eukaryotic cells that sequesters cytoplasmic components within double-membrane vesicles that mature into autophagosomes, which eventually fuse with lysosomes or vacuoles to induce degradation and recycling of their cargo [13,14,15]. This process plays an important role in a variety of physiological functions, such as energy metabolism, organelle turnover, and growth regulation, and is also implicated in diseases such as cancer, cardiomyopathy, neurodegenerative disorders, and autoimmune pathologies [13,15,16]. The regulation of macroautophagy involves a complex interplay of kinases, phosphatases, and GTPases, along with specific protein machinery that drives the formation and consumption of intermediates in the pathway [15,16]. Chaperone-mediated autophagy [CMA] is a selective lysosomal pathway that is important for maintaining cellular proteostasis by degrading specific cytosolic proteins, and LAMP-2A and Hsc70 play important roles in substrate recognition and translocation [17,18,19,20]. CMA is involved in regulating cellular energy balance through amino acid recycling and plays a regulatory role in processes such as lipid and glucose metabolism, cell cycle, and DNA repair [18]. Dysfunctional CMA is associated with a variety of human pathologies, such as aging and metabolic disorders, neurodegeneration, cancer, and diabetes [18]. Microautophagy is the process of repairing damaged lysosomal membranes through a mechanism called “microlithophagy”, which involves the mobilization of key regulators, such as the AGC kinase STK38 and lipidation of ATG8, especially the GABARAP subfamily, for ESCRT assembly [21]. Studies have shown that ATG and ESCRT proteins play important roles in different types of microautophagy, emphasizing the importance of these molecular mechanisms in maintaining cellular homeostasis [22,23].

The process of autophagy is generally divided into the following steps: induction, core complex formation, expansion, maturation, degradation, and recycling. When a cell forms an autophagosome and fuses it with a lysosome, digestive enzymes inside the lysosome break down the contents of the autophagosome to be used for synthesis of new cellular components or energy generation. Autophagy plays a complex role in tumorigenesis, health, and disease, acting as a double-edged sword, inhibiting or promoting tumorigenesis and being associated with a variety of disease processes [24,25,26,27]. Mutations in autophagy-related genes have been identified in human diseases such as neurodegenerative disorders, infectious diseases, and cancer, emphasizing the importance of these genes in pathological conditions [27]. In a previous study, we studied the autophagy molecule ATG7 and autophagy in triple breast cancer [28]. Autophagy plays an important role in many cancers, not only breast cancer, but also GBM, bladder cancer, lung cancer, and others [28,29,30].

Treatments for GBMs include the traditional chemotherapy drug TMZ and radiation therapy. TMZ is a guanine alkylating agent, which reacts at the N-7 or O-6 position of guanine to damage DNA [31,32], but resistance develops and additional resection or radiation therapy do not completely eradicate the tumor. Furthermore, GSCs remaining in brain tissue after radiation and chemotherapy become chemoresistant, leading to accelerated tumor growth [33]. A recent study showed that inhibiting O-GlcNAcylation in GBMs decreases cell viability, inhibits autophagy, and increases sensitivity to the chemotherapeutic agent TMZ [34,35]. O-GlcNAcylation plays an important role in regulating cell proliferation, autophagy, and apoptosis, with excess O-GlcNAcylation promoting increased cell proliferation through activation of autophagy, while low O-GlcNAcylation inhibits autophagy, induces apoptosis, and reduces proliferation [34]. In addition, targeting OGT, the enzyme responsible for O-GlcNAcylation, has been shown to sensitize GBM cells to TMZ treatment without affecting non-tumor cells, which may provide a promising therapeutic strategy for treating this aggressive brain tumor.

Not only is autophagy induced by TMZ treatment considered a chemoresistance mechanism because it mostly acts as a survival and protective mechanism, but autophagy is also induced by TMZ treatment [36], which makes the remaining GSCs chemoresistant, and autophagy affects GSC tumor growth [37]. This demonstrates the importance of understanding the mechanisms of resistance and TMZ treatment-induced autophagy in advancing the treatment of GBM. Autophagy induced by TMZ treatment is considered a chemoresistance mechanism, mostly acting as a survival and protective mechanism. Autophagy is induced upon TMZ treatment [36], making the remaining GSCs chemoresistant, and autophagy affects GSC tumor growth [37]. In addition, CCNU (lomustine) is used as a second-line treatment for patients with recurrent GBM, but low antitumor responses have been reported [38,39]. These findings emphasize the importance of understanding the mechanisms of resistance and TMZ treatment-induced autophagy in the advancement of GBM therapy.

## 2. The Major Molecular Pathways in GMB

### 2.1. Overexpression of EGFR and EGFRvIII and Their Role in GBM Progression

The EGFR is an extracellular protein, also known as ErbB or HER1, and is a transmembrane protein comprising 1186 amino acids. EGFR is overexpressed and overactivated in GBMs and many other malignancies, leading to the exploration of therapeutic strategies that target EGFR. Mutations and overexpression of EGFR also occur in more than 57% of GBMs in TCGA data [40]. Amplification of EGFR is associated with genomic rearrangements observed near the 3′ end of the gene, resulting in deletions encoding portions of the N-terminal and C-terminal tails, and these EGFR gene deletions have evolved to increase oncogenic potential [41]. One of the most common variants of EGFR is a mutation that results in a deletion of exons 2–7 of EGFRvIII, rendering the receptor unable to bind to known ligands. It is expressed in 20–78% of breast tumor tissue, and high expression of EGFRvIII mRNA has been observed in approximately 68% of primary invasive breast cancers and found in 16–39% of NSCLC [42,43]. It has also been reported that in GBM, the overall prevalence of EGFRvIII in patients with EGFR gene amplification ranges from 50% to 60% [44] and contributes to tumor stem cell maintenance [45]. In one study, EGFRvIII induced the secretion of pigment epithelium-derived factor (PEDF), which activated notch signaling and increased Sox2 expression in GSCs, promoting self-renewal and infiltration. Inhibition of PEDF reduced GSC self-renewal and increased survival [46]. The importance of EGFRvIII in other tumors beyond the brain warrants further study.

### 2.2. Activation of the PI3K/AKT/mTOR Signaling Pathway: A Central Role in GBM

PI3K/AKT/mTOR, a well-known intracellular signaling pathway, is activated in nearly 90% of all GBMs and is involved in many functions, including cell proliferation, growth, and migration. phosphatidylinositol 3-kinase class (PI3K) is activated by EGF, which directly affects the PI3K pathway when EGFR is phosphorylated, and when RTKs are phosphorylated [47,48] (Figure 1). These pathways are PI3K, AKT, and mammalian target of rapamycin (mTOR), and when activated upon phosphorylation, they alter signaling for cell survival, growth, and proliferation, as well as induce deletion and silencing mutations of PTEN. PI3K, a lipid kinase, also acts within RTK and G protein-coupled receptor (GPCR) systems that generate PIP2 into PIP3. In the context of neurons, it is activated primarily through RTKs and plays a role in the development of survival synapses and critical insulin signaling [49,50]. A complex balance between PIP2 and PIP3 is required to regulate and maintain many cellular processes in the nervous system [51]. This activated PIP3 activates PIP3-dependent serine–threonine kinases, such as PDK1 and PDK2. PDKs have been reported to be located in cholesterol-rich membranes (Figure 1). Through this access, PDKs phosphorylate AKT at T308 to stabilize and activate it. mTORC1 is composed of mTOR, Raptor, and several complexes, and primarily regulates cell growth and metabolism by integrating signals from multiple growth factors, energy supplies, and nutrients.

## 3. Autophagy

Autophagy is known to maintain cellular health by essentially removing damaged proteins and organelles. It occurs in five phases: induction, core complex formation, expansion, maturation, and degradation and recycling. In the induction phase, the pathway of mTOR plays the most important role. mTOR acts as an inhibitor of autophagy and is activated when nutrients are abundant. On the other hand, nutrient deficiency, energy deficiency, and oxidative stress inhibit mTOR and induce autophagy. AMPK senses energy status, inhibits mTOR to induce autophagy, and ULK1/2 activates ULK1/2 when mTOR is inhibited to promote the early stages of autophagy [52]. After induction, the core complex formation step involves the formation of autophagosome precursors. This monomer requires several autophagy-related complexes, with Beclin-1 playing an important role in the initial phase and forming a complex with PI3K-III, followed by ATG9 and ATG14L in complex formation (Figure 2). In the expansion phase, LC3-I, a bilayer membrane-forming monomer, is converted into LC3-II by ATG7 and ATG13 in its cytoplasmic form, and LC3-II binds to the membrane of the autophagosome and helps the autophagosome to expand [53] (Figure 2). The complex of ATG12-ATG5-ATG16L1 plays a critical role in the expansion and formation of the autophagosome membrane [53]. In the mature stage, the autophagosome is fully formed and fuses with lysosomes to form autophagosomes. Once the autophagosome fuses with the lysosome, digestive enzymes are transported into the autophagosome. Finally, in the degradation and recycling step, the material inside the autophagosome is broken down by lysosomal digestive enzymes [52] (Figure 2). The degraded products are recycled within the cell and used in the synthesis of new proteins, lipids, and other cellular components. In this way, autophagy is an important process for removing and recycling unwanted materials within the cell.

### 3.1. Autophagy in Cancer

Autophagy is an important process within cells that maintains cellular homeostasis by disposing of unwanted proteins and damaged mitochondria. It has a dual role in preventing DNA damage by scavenging reactive oxygen species (ROS) and inhibiting cancer cell growth by inducing apoptosis [54]. ROS and reactive nitrogen species (RNS) cause DNA damage, which contributes to the progression of inflammation-related diseases and cancer. Chronic inflammation triggers the release of ROS and RNS, resulting in DNA damage [55], which can lead to mutations and genomic instability. Inflammatory processes trigger oxidative/nitrosative stress, which generates excess ROS, RNS, and DNA-reactive aldehydes, contributing to DNA damage in cancer-prone diseases [56]. At the same time, however, autophagy can contribute to the survival and growth of cancer cells under multiple stress conditions and increase their resistance to anticancer therapies.

Although autophagy can contribute to the survival and growth of cancer cells and increase their resistance to anticancer therapies, it can also cure cancer by inducing autophagy [57,58], and conversely, it can inhibit tumor growth and metastasis by inhibiting autophagy [57]. Several studies have shown that autophagy inhibition inhibits the metastasis and growth of melanoma [57], breast cancer [58], colon cancer [59], and brain tumors [60].

In cancer cells, autophagy can delay cell death upon DNA damage by maintaining the energy needed to support the DNA repair process [59,60]. Conversely, in cells where DNA is not repaired and apoptosis is defective, autophagy induced by DNA damage has been reported to contribute to cell death, acting as a tumor suppressor process [61]

### 3.2. Role of Autophagy in GBM

Autophagy is an important cell-autonomous process that helps cells survive under stressful conditions. Many cancers have high expression of autophagy. Autophagy is also expressed in GBM patients, but is particularly high in high-grade patients [62]. Autophagy aids cell survival by relieving stress within arrest and removing damaged organelle proteins, which can increase the survival and treatment resistance of GBM cells. Treatments for GBM include surgery, radiation therapy, and chemotherapy, and these treatments often stress cancer cells. In addition, autophagy is highly dependent on the tumor and its stroma, especially in nutrient-limited microenvironments, which can support tumor growth or help cells survive and grow [63]. GBM is one of the most hypoxic tumors in the central nervous system. Hypoxia-induced resistance to TMZ treatment inhibits apoptosis through the miR-26A/BAD/bax axis [64,65]. Hypoxia may induce TMZ resistance by triggering a protective response in mitochondria through HIF-1α-mediated upregulation of miR-26a [64].

In this regard, DNA Damage-Regulated Autophagy Modulator 1 (DRAM1) plays an important role in regulating autophagy and apoptosis and influences the pathogenesis of diseases such as inflammatory bowel disease (IBD) [66]. In GBM, autophagy regulation is an important factor in cell death sensitization and proliferation control [67]. One study showed that BRZ, a novel acetamide, is a potent cytostatic autophagy modulator and showed promising results in reducing neuroinflammation and inhibiting GBM progression [68]. Thus, DRAM1 and other autophagy modulators, such as BRZ, can potentially affect lysosomal function and autophagic flux stability in GBM cells, providing an avenue for novel therapeutic interventions to halt GBM progression and improve patient outcomes. Under these stressful conditions, blockade of the autophagy mechanism is targeted to inhibit the growth of cancer cells [69,70,71,72]. The autophagy-related Beclin-1 molecule is often overexpressed in GBM cells, which may help cancer cells survive through autophagy activation [73].

## 4. TMZ’s Mechanism of Action

A limitation of GBMs is that many drugs do not effectively penetrate into the brain, as the tight junctions of the mucosal cells and the blood–brain barrier (BBB) restrict the passage of molecules larger than 400 Da. However, TMZ has a molecular size of 194 Da, which is hydrophilic enough to effectively penetrate the central nervous system [74]. TMZ is known to be an important drug in the treatment of GBM, but at least 50% of patients undergoing treatment do not respond to TMZ. Nitrosourea, another treatment option, is also used in high-grade malignant brain tumors [75]. Both of these drugs have a mechanism of action that targets DNA and kills cancer cells. While the use of temozolomide and nitrosoureas is centered on damaging the DNA of cancer cells to kill them, there are many patients who do not respond to these drugs. O6-methylguanine-DNA methyltransferase, or MGMT, is a DNA repair enzyme that removes methyl groups attached to O6-methylguanine (damaged guanine) residues and reverts them to normal guanine. GBM is also an inherently heterogeneous tumor and one study looked at changes in MGMT promoter methylation status, dividing the study into three groups: methylated, unmethylated, and methylated to unmethylated, and observed that methylation status changed in recurrent GBMs after standard treatment [76]. In fact, patients with methylated MGMT promoter GBMs had higher PFS at 10 and 4 months, respectively, compared to unmethylated MGMT promoter unmethylated patients [77]. One study found that tumors with high MGMT expression were more resistant to TMZ, and that tumors with MGMT promoters were more sensitive [78]. In some MGMT unmethylated tumors, TMZ treatment induced MGMT expression, thus increasing resistance. MGMT expression is generally inhibited in tumors by CpG methylation within MGMT promoters, and tumor MGMT hypomethylation is associated with TMZ resistance in GBM patients [79,80,81,82]. Furthermore, TMZ increases reactive oxygen species (ROS), and high levels of ROS may contribute to tumor progression and therapeutic resistance. The combination therapy of metformin (MET) and TMZ showed the effect of increasing the Bax/Bcl-2 ratio, reducing ROS production and regulating cell death [78]. Moreover, TMZ increases ROS [83,84], and high levels of ROS may contribute to tumor progression and treatment resistance. Combination treatment of metformin (MET) and TMZ increased the ratio of Bax/Bcl-2, which decreased the production of ROS and, thus, regulated cell death [83]. Consequently, MGMT methylation status has an important impact on the responsiveness of GBMs to TMZ and its resistance to treatment.

## 5. TMZ Treatment Resistance Pathways

### 5.1. EGFR, EGFRvIII

Resistance to TMZ is one of the hurdles to overcome in the treatment of GBM, and dysregulation of certain molecules or pathways that lead to resistance contributes to TMZ resistance [85]. When DNA is damaged after treatment with TMZ, an increase in reactive oxygen species (ROS) is observed [86] (Figure 3B). Moderate ROS serve as important signaling molecules within cells, but excessive ROS can cause drug resistance as EGFRs, such as EGFRvIII, are recruited to protect cells from ROS-induced damage [87,88]. These EGFRs, such as EGFAvIII, induce activation of the PI3K/AKT/ERK cascade, which leads to GSC enrichment, thereby enhancing resistance to TMZ drugs [89] (Figure 3B). The most commonly seen mutation in EGFRs is EGFRvIII, and amplification of EGFR is associated with hypoxia-induced resistance to therapy and resistance to microenvironmental factors [90,91,92,93,94]. It is well established that autophagy is rapidly activated in hypoxia [95,96]. In one study, expression of EGFRvIII was associated with resistance to treatment and accelerated tumor growth in GBM patients, and EGFRvIII tumors had faster tumor growth and regrowth after radiation therapy [97]. Furthermore, the autophagy inhibitor CQ improved survival in all GBM patients, but had a greater effect in EGFRvIII patients [97].

### 5.2. PI3K/AKT/mTOR

Aberrant signaling due to amplification of EGFR or mutations in EGFRvIII interacts with the PI3K/AKT/mTOR pathway to promote chemoresistance and survival in GBMs [98] (Figure 3A). Aberrant expression of PI3K/AKT/mTOR is also recognized as a key regulator of autophagy. Inhibition of the PI3K/AKT/mTOR pathway is known to promote autophagy, and autophagy is an important mechanism to protect cells from TMZ-induced apoptosis [36,99]. Therefore, previous clinical studies using individual inhibitors of AKT and mTOR did not yield significant results [100,101,102]. However, in a recent study, NVP-BEZ235 (BEZ235), an oral PI3K/mTOR dual inhibitor, inhibited autophagy and increased cell death in RCC cells and breast cancer [103,104]. Treatment with 3-MA, a PI3K inhibitor, reduces TMZ-induced cytotoxicity by inhibiting autophagy at an early stage [105,106], but bafilomycin A1, a late-stage inhibitor, increases the permeability of both mitochondrial and lysosomal membranes, activating caspase-3 to induce apoptosis [106]. These roles of autophagy can cause death or cancer survival, depending on the type and stage of cancer or the molecular mechanisms of chemoresistance, and, therefore, the regulation of autophagy needs to be investigated.

### 5.3. GSC

GSCs play a role in the initial formation and development of tumors in GBM, driving the division and proliferation of tumor cells and promoting their invasion and metastasis to surrounding tissues. In addition, GSCs have high autophagy compared to other parental monolayer cells because they must undergo frequent metabolic adaptations to survive in an environment that is low in oxygen and nutrients and rich in waste and necrotic tissue [107,108]. Treatment of GSCs in GBMs remains an obstacle due to their drug resistance. The dependence of CSCs on autophagy has also been implicated in many other cancers, including breast cancer, pancreatic cancer, and ovarian cancer [109,110,111]. GSCs can initiate tumors to become chemoresistant [112], radioresistant, and recurrent [33] (Figure 3B). TMZ is also known to induce tumor recurrence from non-GSCs to GSCs [113]. TMZ treatment induced autophagy and prevented the loss of MMPs in the removal of unhealthy proteins, mitochondria, and stressors, and maintained cells in a healthy state despite elevated ROS levels. Furthermore, the survival, metabolic capacity, and size of GSCs were significantly increased, confirming that the use of autophagy inhibitors 3-MA and CQ not only increased the toxic effects of TMZ but also inhibited the pool of GSCs [108] (Figure 3B). The role of autophagy depends on the type and stage of cancer, or the molecular mechanism of chemical resistance, and low levels of DNA damage result in protective effects, while high levels of damage in autophagy contribute to toxicity, which requires further research.

## 6. Autophagy-Regulating Drugs

### 6.1. Bevacizumab

Bevacizumab was approved for medical use in the United States in 2004 and is used to treat cancer and certain eye diseases. For cancer, it is injected into a vein and is used for several cancers, including colorectal cancer, lung cancer, and glioblastoma. Bevacizumab is an angiogenesis inhibitor and belongs to a class of drugs called monoclonal antibodies. Treatment of GBM patients with bevacizumab in combination with radiation and TMZ has been shown to increase the resolution rate of thromboembolic side effects, suggesting a novel approach [114], although in chemotherapy-resistant GBM, bevacizumab alone has been shown to alter the cellular environment by inhibiting angiogenesis in GBM xenografts, triggering a stress response that increases autophagy, which promotes tumor cell growth and resistance to neoangiogenic therapy. In combination with the autophagy inhibitor CQ, treatment interferes with tumor growth, suggesting a novel mechanism to overcome resistance to anti-angiogenic therapy for GBM [115] (Table 1).

### 6.2. Resveratrol

Resveratrol, a natural compound, has been studied for its beneficial effects in cancer. In one study, TMZ induced autophagy and apoptosis in GBM, resulting in a significant increase in autophagy due to increased ROS/ERK. Here, resveratrol inhibited autophagy, causing TMZ to increase apoptosis, suggesting a combination treatment of TMZ and resveratrol [86] (Table 1).

### 6.3. 3-MA

3-Methyladenine (3-MA) is an inhibitor of PI3K, one of the drugs that regulates the activation of mTOR, a key regulator of autophagy. When used in combination with TMZ in GBM, it increases U373-MG cell survival from 37% to 52%, inhibits the antitumor effects of TMZ, and may inhibit cytotoxicity [106] (Table 1).

### 6.4. Bafilomycin A1

Bafilomycin A1 is an inhibitor of vacuolar H+ ATPase, which inhibits the role of late autophagy by preventing its binding to the lysosome. It sensitizes tumor cells to TMZ by inducing apoptosis through activation of caspase-3, and the cytotoxicity of TMZ is enhanced, which may increase survival by inducing apoptosis [106] (Table 1).

### 6.5. CQ

CQ is known as an antimalarial drug, but it is also an autophagy inhibitor by directly entering lysosomes and neutralizing Ph to disrupt the fusion of lysosomes and autophagosomes. Treatment with CQ in GBMs can enhance the anti-cancer effect of TMZ by blocking the autophagy process, causing apoptosis, which inhibits the growth of cancer cells and increases the effectiveness of the treatment [116] (Table 1).

### 6.6. HCQ

HCQ is also known as an antimalarial drug, but it is a derivative of CQ with a hydroxyl group added to the CQ molecule. HCQ is less toxic in the body than CQ and generally has fewer side effects with long-term use. However, in one clinical trial, patients receiving oral HCQ (200 mg to 800 mg) with TMZ at the same time as radiation failed to achieve autophagy and had dose-limiting toxicity and minimal efficacy [117,118] (Table 1).

### 6.7. Lys05

Lys05 is an analog of the lysosomal inhibitor CQ, which has a much stronger autophagy-inhibitory effect and demonstrated antiglioma activity by reducing cell survival and growth in U251 and LN229 glioma cells. Lys05 induced LMP and mitochondrial depolarization, and increased radiosensitivity; therefore, as an autophagy inhibitor and LMP inducer, Lys05 may be a promising compound for GBM treatment [119] (Table 1).

## 7. Conclusions

Exactly what role autophagy plays in cancer, whether it promotes or inhibits tumor growth, is still under investigation. EGFR-TKIs and other drugs are used for targeted therapy of EGFR and EGFRvIII, but are often limited by resistance. EGFR is highly expressed in NSCLCs, and drugs such as lapatinib have been effective in NSCLCs but less so in GBMs. This is because EGFR mutations occur in the kinase domain in NSCLCs, but are primarily extracellular in GBMs, which is a limitation in GBMs and may also limit treatment due to the rapid growth of autophagy in hypoxia [78]. PI3K/AKT/mTOR signaling affects cell growth and metastasis and is also important in the regulation of autophagy. While the mTOR inhibitor IDELALISIB, which is currently approved for the treatment of cancer with specific pharmacologic agents, has not been successful [79], inhibitors that inhibit both PI3K/mTOR have shown good results by inducing apoptosis as well as inhibiting autophagy [57].

In response to TMZ-induced DNA damage, autophagy, senescence, and apoptosis are induced [120,121]. Autophagy acts as a survival mechanism, promoting senescence against apoptosis, and inhibition of autophagy after TMZ treatment increases the level of apoptosis [36]. Induction of senescence by O6Meg is a late response that occurs at a higher level than apoptosis and is characterized by unrepaired DNA double-strand breaks and sustained activation of the DNA damage response [122]. Furthermore, the interaction between autophagy and senescence in response to TMZ is complex, with autophagy negatively correlating with senescence markers at the population level, whereas no such correlation was observed at the single-cell level [123].

In addition, resistance to treatment with surgical radiation and chemotherapy, which is still limited in GBMs, is still a problem due to recurrence and poor prognosis. Treatment in GBMs includes chemotherapy, radiation, and resection. However, recurrence and poor prognosis are very problematic, so two treatments are often combined, such as chemotherapy and radiation or resection and chemotherapy. Treatment with TMZ, a chemotherapeutic agent, induces autophagy for several reasons, including increased ER stress [124], damage to the mitochondria, and increased ROS. Tumors and cells remaining after such TMZ treatment receive energy from autophagy to promote growth and proliferation, which need to be regulated. In addition, tumors and GSCs that remain after resection or after radiation therapy may be chemoresistant, which means that GSCs rely on autophagy to grow because they require low oxygen and nutrients, and autophagy provides them with a lot of nutrients, making them resistant.

In conclusion, we show that targeting the autophagy pathway is a promising approach to enhance the efficacy of existing GBM therapies and overcome treatment resistance through inhibition of PI3K/AKT/mTOR signaling. Furthermore, autophagy inhibitors are essential for increasing the toxicity of TMZ and overcoming the resistance of GSCs. We emphasize the need for further research to understand the specific mechanisms of GBM resistance and develop targeted therapies to effectively inhibit it.

## Figures and Tables

**Figure 1 cells-13-01332-f001:**
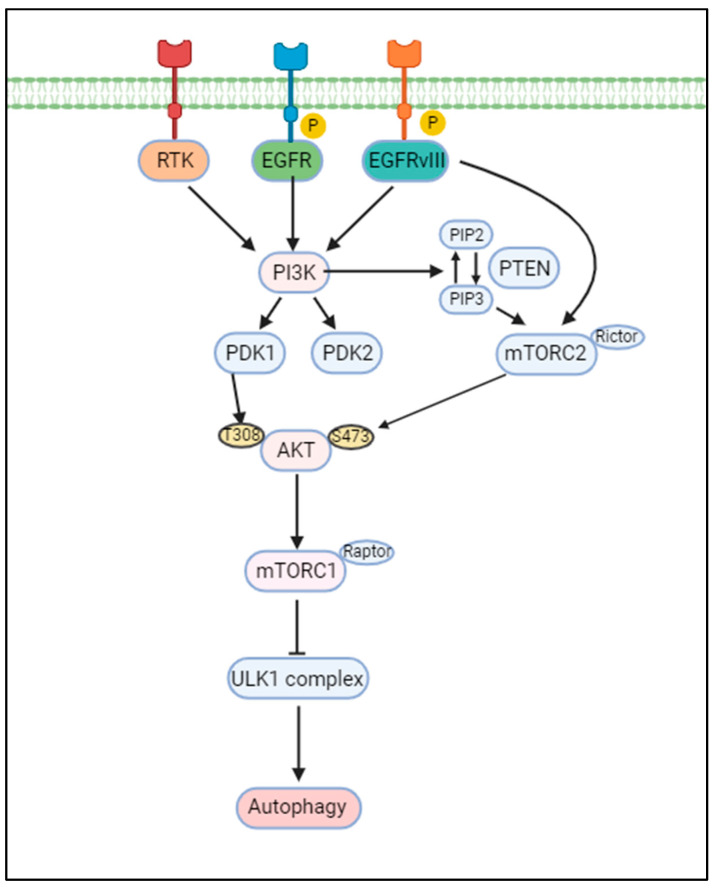
Receptor tyrosine kinases (RTKs), epidermal growth factor receptor (EGFR), and its variant EGFRvIII activate the PI3K/AKT signaling pathway, subsequently influencing mTOR activity. Activation of RTKs, EGFR, and EGFRvIII leads to the phosphorylation and activation of PI3K, which converts PIP2 to PIP3 [18]. This conversion recruits and activates AKT, which in turn phosphorylates and activates mTOR. Activated mTOR functions as a major inhibitor of autophagy by phosphorylating and inhibiting key autophagy-initiating proteins, such as ULK1/2 [52]. This inhibition prevents the formation of autophagosomes and, subsequently, blocks the autophagic process, allowing for the accumulation of damaged proteins and organelles within the cell. mTOR pathway: Inhibition of autophagy occurs through the mTOR pathway, which is activated by abundant nutrients. Under conditions of nutrient deficiency, energy deficiency, and oxidative stress, mTOR is inhibited, leading to the induction of autophagy.

**Figure 2 cells-13-01332-f002:**
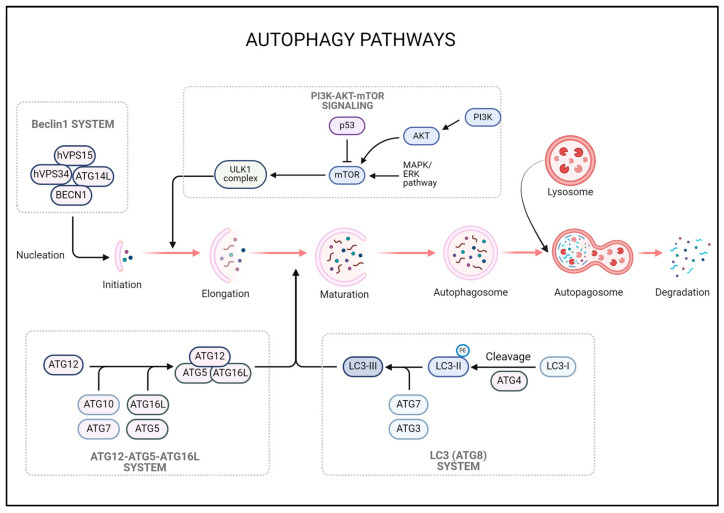
Beclin-1 forms a complex with PI3K-III, ATG9, and ATG14 to initiate autophagosome formation. LC3-I is converted to LC3-II by ATG7 and ATG13 [53]. LC3-II binds to the autophagosome membrane and helps it expand. The ATG12-ATG5-ATG16L1 complex is essential for the extension and formation of the autophagosome membrane and promotes growth [53]. Mature autophagosomes fuse with lysosomes to form autolysosomes. Lysosomal digestive enzymes are transported into the autophagosome. Lysosome and autophagosome fuse to break down damaged organelles and dysfunctional substances to create new proteins and various substances.

**Figure 3 cells-13-01332-f003:**
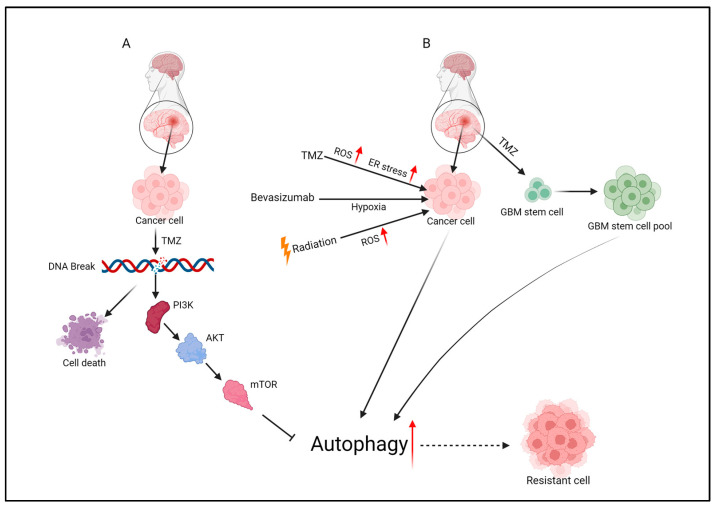
Temozolomide (TMZ) treatment leads to an increase in reactive oxygen species (ROS) and endoplasmic reticulum (ER) stress in cancer cells [86] (**A**,**B**). These stress factors induce autophagy as a survival mechanism in cancer cells. Bevacizumab treatment causes an increase in hypoxia within the tumor microenvironment. Hypoxia acts as a trigger for autophagy, further promoting the survival of cancer cells under stress conditions. Radiation therapy results in an increase in ROS levels. The heightened ROS levels stimulate autophagy, enabling cancer cells to cope with the oxidative damage induced by radiation. Post-TMZ surgery, following TMZ treatment and surgical intervention, residual glioma stem cells (GSCs) survive. These GSCs, now under selective pressure, form resistant cell populations, leading to the development of drug resistance. This figure illustrates the complex interplay between various treatments (TMZ, bevacizumab, radiation) and their impact on the induction of autophagy in cancer cells. It highlights the role of autophagy in promoting cancer cell survival and the subsequent emergence of drug-resistant cells.

**Table 1 cells-13-01332-t001:** Combination Therapies with Temozolomide and Their Effects on Glioma Treatment.

	Combination Therapy	Drug	In Vitro	In Vivo	Human	Result	Reference
TMZ	RT	Bevacizumab			O	A higher proportion of patients in the bevacizumab group completed 6 cycles of maintenance temozolomide (64.6%) compared to the placebo group (36.9%).	[115]
		Resveratrol		O		Mice treated with the combination therapy also showed long-term survival compared to mice treated with TMZ alone.	[86]
		3-MA	O			Increased cytotoxic effects on glioma cells when 3-MA was combined with TMZ.	[106]
		Bafilomycin A1	O			Treatment of glioma cells with both Baf A1 and TMZ has been shown to significantly reduce cell viability compared to TMZ alone.	[106]
		CQ	O	O		CQ enhances the cytotoxicity of TMZ in glioma cells by blocking autophagy, which in turn increases ER stress and promotes apoptosis.	[116]
	RT	HCQ			O	In the phase 2 cohort of 76 patients, the median survival was 15.6 months. The 12-, 18-, and 24-month survival rates were 70%, 36%, and 25%, respectively.	[117,118]
	RT	Lys05	O			Lys05 exhibits cytotoxic effects in glioma U251 and LN229 cells by decreasing cell viability and reducing cell growth.	[119]

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
