# Peer review of "Importance of Autophagy Regulation in Glioblastoma with Temozolomide Resistance"

_cells, 2024, doi:10.3390/cells13161332_

Round 1

Reviewer 1 Report

Comments and Suggestions for Authors

This manuscript by Hwang et al reviews the role of autophagy in glioblastoma drug resistance, with focus on temozolomide.

There are some repetitions; the manuscript needs careful revision.

Specific comments:

-       Abstract: line 13. 47.4% of all cases. Which cases? All cancers or just malignant brain cancer?

-       Line 50: delete “etc” or be more specific.

-       2nd line therapy for glioblastoma is CCNU (lomustine). This drug is even not mentioned.

-       Line 51,52:  add guanine to the alkylation positions

-       Line 49-52: add a relevant reference, e.g. a recent review [1].

-       Line 55: add the missing reference

-       Line 57: The finding that TMZ induces autophagy (and the mechanism behind) was actually shown for the first time in this paper and, therefore, please add the reference: Knishnik et al. [2]

-       Line 84: delete “within all brain tumors”

-       Line 90: delete “and”

-       Add a citation for the statement that autophagy can prevent DNA damage by scavenging ROS. Explain more precisely. How can this be accomplished?

-       Line 165: hypoxia from TMZ treatment: where was this shown? Does TMZ cause hypoxia?

-       Chapter 3. This is a quite superficial description of what is known as to the molecular mechanism of TMZ. Also, there is some repetition with the introduction. I suggest adding some more references. A recent review is recommended (see ref. 1 or [3]).

-       Line 189: TMZ is a methylating agent, and MGMT in a therapeutic dose range can eliminate all toxic effects. I have doubts that TMZ induces significant ROS and that ROS in therapeutic doses (<50 µM) plays a role in genotoxicity and autophagy. Show the evidence. The reference 37 does not show evidence; it is not experimental work.

-       Line 191-192: The sentence “Although methylations can be predicted by MGMT, most GBMs have mutations in IDH” is nonsense. How can methylations be predicted by a repair enzyme? Do authors confuse promoter methylation with MGMT expression? And what has this to do with IDH1?

-       In this chapter authors refer to papers (ref. 38, 39) that are not anymore topical. MGMT promoter methylation is now current standard in GBM diagnostics. Rewrite and refer to more recent papers (see also comment below).

-       Line 192 pp: IDH1 mutated gliomas are grade 3 astrocytoma. They are not generally silenced for MGMT through MGMT promoter hypermethylation. The sentences line 192 -197 are confuse. They must be rewritten and the facts should be better explained in their own words (and not just copied from other reviews). This, explain please, what CpG methylation actually means? What does the statement mean: “limiting at the MGMT promoter”?

-       Line 204: Fig. 3B: There is no Fig. 3B nor Fig. 3A.

-       Line 203: Reference37: Please check whether this are therapy relevant doses used here.

-       Line 221: Autophagy clearly protects against TMZ-induced apoptosis, which was shown by Knizhnik et al. [2]. Again, at very high dose levels the opposite may be observed, but this is not relevant for therapy.

-       Line 260: It seems there is a general rule: Low DNA damage level results in autophagy that is protective, high damage level in autophagy that contributesto toxicity.

-       Line 284: percentage of mice or human patients?

-       Table 1: Add a separate column for the references.

-       Line 326-327: An important endpoint of TMZ is cellular senescence [4]. Does autophagy protect against senescence? See again the paper by Knizhnik [2]. I recomment to add a small chapter addressing this important issue.

Citations:

1.             Kaina, B.; Christmann, M., DNA repair in personalized brain cancer therapy with temozolomide and nitrosoureas. DNA Repair (Amst) 2019, 78, 128-141.

2.             Knizhnik, A. V.; Roos, W. P.; Nikolova, T.; Quiros, S.; Tomaszowski, K. H.; Christmann, M.; Kaina, B., Survival and death strategies in glioma cells: autophagy, senescence and apoptosis triggered by a single type of temozolomide-induced DNA damage. PLoS One 2013, 8, (1), e55665.

3.             Tomar, M. S.; Kumar, A.; Srivastava, C.; Shrivastava, A., Elucidating the mechanisms of Temozolomide resistance in gliomas and the strategies to overcome the resistance. Biochimica et biophysica acta. Reviews on cancer 2021, 1876, (2), 188616.

4.             Beltzig, L.; Schwarzenbach, C.; Leukel, P.; Frauenknecht, K. B. M.; Sommer, C.; Tancredi, A.; Hegi, M. E.; Christmann, M.; Kaina, B., Senescence Is the Main Trait Induced by Temozolomide in Glioblastoma Cells. Cancers 2022, 14, (9).

Comments on the Quality of English Language

No comment

Author Response

Please see attached the response to the reviewer.

Reviewer 2 Report

Comments and Suggestions for Authors

There is basic level review on GBM-autophagy interactions  in development of anticancer drug’s resistance or susceptibility. Basic pathways of intracellular signal transduction generating pro or anti-survival modifications are described. No substantive inconsistences were found.  The article may be useful for starting deeper studies on anti-glioblastoma mechanisms. One should carefully check whether some data from peripheral cancer apply to GBM. Final conclusion is inconclusive. Such statements should be avoided.

Author Response

Please find attached the response to the comments.

Reviewer 3 Report

Comments and Suggestions for Authors

In the Review Article “Importance of autophagy regulation in Glioblastoma with Temozolomide resistance”, Hwang and colleagues discussed aspects of how autophagy influence glioblastoma biology, including chemotherapy resistance. The topic is interesting. However, the authors wrote a superficial manuscript that does not capitulate the current status of the field since key citations are not included and discussed. Major revisions must be performed:

Introduction – The introduction is superficial and not well supported by citations. The first 3 paragraphs comprehend the importance of glioblastoma for patients diagnosed, epidemiological data and an introductory discussion and definition of autophagy. These 3 major points for the whole review have 3 citations and was written in 20 lines. Expand and discuss in detail: What is the epidemiology of glioblastoma compared to other CNS tumours? How it is diagnosed, what are the markers? What characterizes glioblastoma? How is autophagy defined? What are the subtypes (macroautophagy, chaperone-mediated, microautophagy)? Despite parts of these questions are answered in latter sections, I feel that those are introductory, key concepts for the rest of the review, and thus should be placed in the introduction section. As a suggestion, include 2 topics in the introduction – 1) Glioblastoma biology (or something related); 2) Autophagy, general aspects.

Again, for a review article in two rapidly-growing fields, autophagy and glioblastoma biology, the authors have cited a small amount of literature. This review would greatly improved if important and missing references are included and discussed, such as PMID: 37835434, PMID: 36635405, PMID: 36864290, PMID: 38537746, PMID: 38395990 and many others.

Each of the topics must be improved with higher detail in the discussion and more citations. I do not feel the current pipeline of ideas is interesting. The authors might look at rearrangements for improved clarity for the target audience. For example, putting the signaling pathway section after the discussion about the effects of autophagy for cancer biology and glioblastoma.

Comments on the Quality of English Language

The english is appropriate.

Author Response

(The authors gave the same response as above.)

Round 2

Reviewer 1 Report

Comments and Suggestions for Authors

The revised version is improved.

Having a glance on it, it still needs correction:

Line 12 and line 26: What is Glioblastoma polymorpha? The old terminology was Glioblastoma multiforme, which was replaced by Glioblastoma (GBM). Correct the two lines.

Line 28: Replace the word “it” by “gliomas”.    Note: Gliomas are subdivided, not glioblastomas.

Line 30:  high grade glioma

Line 719: inhibition of apoptosis …increases the level of apoptosis.  Be more careful in your writing!  You presumably mean:  inhibition of autophagy ….increases the level of apoptosis.  This must be corrected.

Comments on the Quality of English Language

ok

Author Response

Point-by-point response to the reviewer comments

Thank you very much for your meticulous review of the manuscript and useful feedback.

There were a lot of corrections and missed parts while writing. Thank you so much again.

I checked the modifications in red.

Reviewer: 1

Reviewer’s comment 1 :

Line 12 and line 26: What is Glioblastoma polymorpha? The old terminology was Glioblastoma multiforme, which was replaced by Glioblastoma (GBM). Correct the two lines.

Author’s response

Glioblastoma polymorphoma (GBM) is the most aggressive and common malignant and CNS tumor, accounting for 47.7% of the total.

- Glioblastoma (GBM) is the most aggressive and common malignant and CNS tumor, accounting for 47.7% of the total.

Glioblastoma multiforme (GBM) is one of the most malignant tumors that occur in the central nervous system (CNS) and one of the cancers with a very poor prognosis.

- Glioblastoma (GBM) is one of the most malignant tumors that occur in the central nervous system (CNS) and one of the cancers with a very poor prognosis.

Reviewer’s comment 2 :

Line 28: Replace the word “it” by “gliomas”.    Note: Gliomas are subdivided, not glioblastomas.

Author’s response 

Organization (WHO) classification, it is divided into low-grade (â… ,â…¡) and high-grade (â…¢,â…£).

- Organization (WHO) classification, gliomas is divided into low-grade (â… ,â…¡) and high-grade (â…¢,â…£).

Reviewer’s comment 3 :

Line 30:  high grade glioma

Author’s response

World Health Organization (WHO) classification, it is divided into low-grade (â… ,â…¡) and high-grade (â…¢,â…£).

- World Health Organization (WHO) classification, gliomas is divided into low-grade (â… ,â…¡) and high-grade (â…¢,â…£).

Reviewer’s comment 4 :

Line 719: inhibition of apoptosis …increases the level of apoptosis.  Be more careful in your writing!  You presumably mean:  inhibition of autophagy ….increases the level of apoptosis.  This must be corrected.

Author’s response

I think it's 399 line, not 719. Thank you.

Autophagy acts as a survival mechanism, promoting senescence against apoptosis, and inhibition of apoptosis after TMZ treatment increases the level of apoptosis (36).

- Autophagy acts as a survival mechanism, promoting senescence against apoptosis, and inhibition of autophagy after TMZ treatment increases the level of apoptosis (36).

I learned a lot from this reviewer, and thank you so much for the feedback.